# DIVERGENCE-INDUCED CONTRASTIVE UNLEARNING VIA DIRECTED REPRESENTATION SHIFTS

## ABSTRACT

Unauthorized data collection has become widespread, raising the need for defenses that prevent exploitation of personal data. Unlearnable Examples (UEs) address this by embedding imperceptible perturbations that preserve visual quality while making data unusable for training. Recent work has shown that contrastive learning can be poisoned to generate UEs, but existing methods lack theoretical grounding and fail to exploit the geometric structure of learned representations. In this work, we present the first principled analysis of contrastive poisoning and reveal why it is effective. Building on this understanding, we propose Divergence-Induced Contrastive Unlearning (DICU), a framework that introduces direction-aware divergence regularization into the poisoning objective. This design amplifies intra-class sparsity, pushes samples beyond class manifold boundaries, and enables free mixing across classes, producing stealthy and robust perturbations. Our approach is especially effective in high class-count settings, reducing linear probing accuracy at significant level.

## 1 INTRODUCTION

The rapid growth of online data has increased concerns about its unauthorized use in training machine learning models. Public datasets have been central to the progress of deep learning, yet their use also raises serious privacy risks (Prabhu & Birhane, 2021; Birhane & Prabhu, 2021). This concern has motivated the development of unlearnable examples (UEs) (Huang et al., 2021; Fowl et al., 2021b), which are designed to make data unusable for training machine learning models. Similar approaches are also referred to as availability attacks (Yu et al., 2022) or indiscriminate poisoning attacks (He et al., 2023) in the literature. These techniques enable users to inject protective noise into their personal data, reducing the risk of unauthorized exploitation.

Existing approaches to unlearning rely on perturbing training data so that models cannot learn meaningful representations (Carlini & Terzis, 2022; Cherepanova et al., 2021; Fowl et al., 2021b). Early methods added error-minimizing noise with surrogate models and produced unlearnable examples at either the sample or class level. These perturbations were fragile and failed under adversarial training. Later work shifted to indiscriminate poisoning, which aims to broadly degrade performance. Most studies were limited to supervised learning with cross-entropy loss (Mei & Zhu, 2015; Muñoz-González et al., 2017), even though contrastive learning can now achieve equal or better performance without labels. Contrastive Poisoning (CP) (He et al., 2023) extended these attacks to contrastive learning by distorting the InfoNCE objective and weakening data augmentation. It also introduced a dual-branch gradient scheme that targets momentum encoders. CP forces augmented poisoned pairs to move closer while the corresponding clean views move apart, creating a new form of unlearning in the contrastive setting. However, it does not address interactions between clean and poisoned pairs, and it does not explain why the attack is so vulnerable. On the other hand, Unlearnable Clusters (UC) (Zhang et al., 2023) advanced the field with cluster-level perturbations. A surrogate model extracted representations, and K-means grouped them into clusters. For each cluster, a generator produced perturbations that shifted samples toward incorrect centers, preventing models from learning valid structures. This made the method label-agnostic and resistant to label-based exploitation. Yet, UC required the generator to be reinitialized for every cluster, which led to high computational cost and poor scalability.

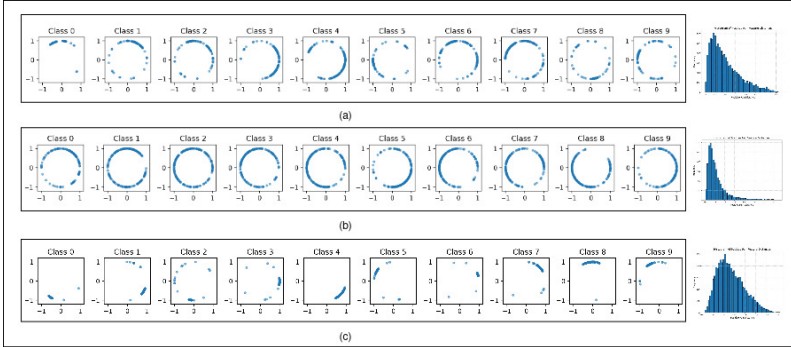

Figure 1: Representation analysis on CIFAR-100 in terms of uniformity and alignment. (a) Contrastive learning with clean samples shows balanced alignment and uniformity. Under contrastive poisoning, (b) poisoned samples achieve higher alignment and maintain uniformity, while (c) clean samples lose uniformity and exhibit larger distances between augmented pairs. This shift indicates increased sparsity in clean class embeddings.

We draw key insights from the representation analysis in Figure 1. Contrastive poisoning yields high alignment and uniformity for poisoned views, but it pushes clean samples apart and reduces their uniformity. This makes clean embeddings sparse within the class manifold and decreases the separation between nearby classes. These findings highlight that managing intra-class structure and embedding sparsity in contrastive frameworks is critical for developing more robust unlearning and defense strategies.

Motivated by these observations, we propose Divergence-Induced Contrastive Unlearning (DICU), a framework that increases intra-class dispersion and encourages mixing across classes. This design is particularly effective in high class-count settings, where it produces a more stealthy attack. Furthermore, our divergence regularization enables samples from one class to blend with those of any other class, enhancing the strength of the poisoning effect. In summary, our main contributions are as follows:

- We propose a stealthy poisoning attack within the contrastive learning framework, where we deliberately increase intra-class distances. By considering scenarios with a larger number of clusters, even a small perturbation can disrupt semantic alignment, leading to semantically entangled yet clustered representations that covertly degrade downstream performance.

- We conduct extensive experiments on datasets with a large number of classes, evaluating multiple models across two different contrastive learning frameworks.

- We extensively evaluate a range of defense mechanisms and observe that, in most cases, the attack remains robust.

- We investigates cross-transferability across diverse datasets and backbone models, revealing the strong robustness of the proposed attack.

## 2 RELATED WORK

**Unlearning Examples.** Unlearnable examples (UEs). UEs are a type of data poisoning attack (Biggio et al., 2012; Biggio & Roli, 2018) designed to prevent models from effectively learning on a protected dataset. Early methods relied on adversarial perturbations to degrade training performance, while recent approaches have focused on improving efficiency, transferability, and generalization across different architectures and datasets. In general, UEs can be generated through a bilevel optimization framework (Huang et al., 2020; Schwarzschild et al., 2021; Shafahi et al., 2018; Zhu et al., 2019) with the aid of a surrogate model (Huang et al., 2021), following a strategy similar to strong data poisoning. Adversarial noise is commonly employed, including methods such as Error-Maximizing

Noise (EMaxN) (Koh & Liang, 2017), Deep-Confuse (Feng et al., 2019), and Adversarial Poisoning (AdvPoison)

**Attacker Objective.**   In contrastive learning, a feature extractor is trained using a self-supervised objective without access to labels (Chen et al., 2020a). The learned representations are typically evaluated on downstream tasks via linear probing (Alain & Bengio, 2017), where a linear classifier is trained on top of the frozen features. The objective of the attacker is to poison the pretraining dataset such that any model trained on it fails to learn transferable or semantically meaningful representations. As a consequence, performance on downstream tasks–particularly under linear probing–deteriorates significantly, reflecting the success of the attack in disrupting representation learning.

**Attacker Capability.**   We consider an indiscriminate poisoning setting where the attacker perturbs the victim's training data to degrade learned representations. The attacker does not control the model architecture, initialization, or training routine, and may or may not know the specific contrastive learning algorithm used–such as SimCLR, MoCo, or BYOL. Following the convention established in prior work (He et al., 2023; Yu et al., 2022), we assume the attacker perturbs 100% of the training data with imperceptible noise constrained by an $\ell_\infty$ norm bound of $\epsilon = \frac{8}{255}$.

## 3   DIVERGENCE-INDUCED CONTRASTIVE UNLEARNING

Contrastive learning is guided by alignment and uniformity objectives, which can be extended to poisoning by training with adversarially crafted samples (see Appendix A.1). We introduce a poisoning framework based on representation-level interference in contrastive learning. Our method, Divergence-induced Contrastive Unlearning (DiCU), breaks intra-class coherence by encouraging divergence among poisoned samples within the same class. In contrast to prior poisoning attacks that mainly induce global class confusion or manipulate decision boundaries in supervised settings (Shafahi et al., 2018; Zhu et al., 2019), DiCU directly targets intra-class structure, fragmenting class manifolds in the contrastive embedding space.

The proposed attack is executed in two stages. In the contrastive poisoning phase, augmented views of poisoned samples are tightly aligned, while clean positives are geometrically displaced across the unit hypersphere. This misalignment distorts the global feature structure, compromising the consistency of learned representations. In the divergence induction phase, we introduce class-wise directional constraints that drive intra-class features apart through controlled angular shifts. This promotes sparsity within class manifolds and weakens inter-cluster boundaries, effectively scattering representations of the same class throughout the embedding space. The combined effect of these two phases significantly degrades representation quality and impairs downstream classification performance. We formalize this behavior using a few geometric definitions.

**Definition 1** (Poisoned Local Consistency). *Let $x_i \in \mathcal{X}$ be a sample from class $c \in \mathcal{C}$. Let $x_i^c$ denote a clean augmented view and $x_i^p$ a poisoned augmented view of the same instance. Let $f_\theta(\cdot)$ be a contrastive poisoned encoder producing representations $z_i^c = f_\theta(x_i^c)$ and $z_i^p = f_\theta(x_i^p)$. A representation space satisfies poisoned local consistency if multiple poisoned augmentations of the same instance remain closely aligned, while their clean counterparts are pushed away in the embedding space. Formally, for poisoned views $x_i^{p(1)}, x_i^{p(2)}$ of a sample $x_i$, $z_i^{p(1)} = f_\theta(x_i^{p(1)})$ and $z_i^{p(2)} = f_\theta(x_i^{p(2)})$ and for the corresponding clean-augmented views $(x_i^{c1}, x_i^{c2})$,*

$$\cos\left(z_i^{p(1)}, z_i^{p(2)}\right) \geq \tau \quad \text{and} \quad \cos\left(z_i^c, z_i^p\right) \leq \epsilon \tag{1}$$

*where $\cos(u, v) = \frac{u \cdot v}{\|u\| \|v\|}$, $\tau$ is a lower bound for poison-poison alignment and $\epsilon$ is an upper bound on clean-poison similarity. This contrastive setup ensures that the poisoned views form a coherent (but misaligned) subspace, effectively detaching them from their clean representation.*

**Definition 2** (Class-Wise Divergence via Targeted Angular Separation). *We define class-wise divergence as the enforcement of a fixed angular displacement between clean and poisoned views of the same input. Let $x_i$ be a sample from class $c \in \mathcal{C}$, and let $x_i^c$, $x_i^p$ be its clean and poisoned augmented views. Let $z_i^c = f_\theta(x_i^c)$ and $z_i^p = f_\theta(x_i^p)$ denote their normalized feature representations. For a*

*prescribed angular margin $\theta_{ref} \in (0, 2\pi)$, we define divergence to hold when:*

$$\cos(\phi) \approx \cos(\theta_{ref}), \;\; where \;\; \cos(\phi) = \frac{\mathbf{z_i^c} \cdot \mathbf{z_i^p}}{\|\mathbf{z_i^c}\| \, \|\mathbf{z_i^p}\|} \tag{2}$$

*i.e., the angle between the clean and poisoned views is driven toward a fixed semantic offset in the latent space. This constraint is enforced to systematically displace poisoned representations from their clean counterparts while maintaining structured orientation.*

To understand how our poisoning mechanism disrupts representation learning beyond local alignment objectives, we examine its global impact on the structure of the learned feature space. While contrastive losses encourage instance-level alignment, robust downstream performance also relies on the formation of coherent class-level manifolds. Our method introduces controlled divergence within each class, which may maintain local consistency but progressively fragments global class structures. To formalize this behavior, we propose the following hypotheses and geometric propositions that characterize the degradation of semantic organization in the latent space under our attack.

### 3.1 BI-LEVEL OPTIMIZATION FOR *DiCU*

After introducing the poisoning framework, we now formulate the optimization of *DiCU* as a bi-level problem. Our goal is to craft poisoned representations that both adhere to local consistency and induce structured intra-class divergence. To achieve this, we alternate between optimizing the contrastive encoder $f_\theta$ and updating the perturbation parameters $\boldsymbol{\delta}$ and direction shifts $\boldsymbol{\beta}$ associated with each class.

Recognizing the complexity of this setup, we follow a staged training strategy: during each training round, we first update the encoder $f_\theta$ using a standard contrastive loss on the perturbed dataset. Then, we optimize the poisoning parameters with respect to a composite loss that enforces directional divergence and sparsity. This bi-level routine is summarized in Algorithm 1.

**Phase 1: Encoder Update.** Given a batch of poisoned samples, we update the encoder parameters $\theta$ using the contrastive loss $\mathcal{L}_{\mathrm{CL}}$, as defined in prior work He et al. (2023). The poisoned views are generated by applying class-specific perturbations:

$$x_i^p = x_i^c + \boldsymbol{\delta}_{y_i}, \;\; \text{where } y_i \text{ denotes class label of sample } x_i. \tag{3}$$

The encoder update minimizes:

$$\theta \leftarrow \theta - \eta_\theta \nabla_\theta \mathcal{L}_{\mathrm{CL}}(f_\theta; \{x_i^p\}_{i=1}^B). \tag{4}$$

**Phase 2: Poison Generator Update.** After the encoder is updated, we refine the poisoning vectors $\boldsymbol{\delta}$ to enforce divergence-aware objectives. This is done by minimizing the directional divergence loss:

$$\mathcal{L}_{DDL} = \frac{1}{|\mathcal{B}|} \sum_{i \in \mathcal{B}} \left( \cos(\phi_i) - \cos(\phi_{\mathrm{ref}, y_i}) \right)^2, \tag{5}$$

where $\cos(\phi_i)$ is computed from the clean and poisoned views of sample $x_i$, and $\phi_{\mathrm{ref}, y_i}$ is a reference direction assigned from a predefined angular set. This loss ensures each poisoned sample is pushed in a distinct direction to induce intra-class fragmentation.

The poisoning parameters are updated to minimize the combined loss:

$$\mathcal{L}_{\mathrm{total}} = \lambda \mathcal{L}_{\mathrm{CL}} + \mathcal{L}_{\mathrm{DDL}}, \tag{6}$$

where $\lambda$ is the regularizer balancing contrastive consistency and sparsity.

The feature extractor is updated using stochastic gradient descent, while the poisoning perturbations are optimized with projected gradient descent (Madry et al., 2018; He et al., 2023) to ensure they remain within the prescribed $\ell_\infty$ norm constraint. All training steps are detailed in Algorithm 1.

## 4 EXPERIMENT

This section presents the experimental setup, which includes datasets, model architectures, poisoning frameworks, baselines, and training details in Section 4.1. The main results and an extensive ablation study are provided in Section 4.2 and Section 4.3. We further include visualizations in **??** to better understand the behavior of DICU.

---

**Algorithm 1** *Divergence-induced Contrastive Unlearning*

---

1: **Input:** Clean dataset $\mathcal{D}_c$; learning rates $\eta_\theta$, $\eta_\delta$; total rounds $T$; updates per round $T_\theta$, $T_\delta$ for feature extractor and perturbations; PGD steps $T_p$; $K$ number of classes.
2: **for** t = 1 to $T$ **do**
3:     **for** $t_\theta = 1$ to $T_\theta$ **do**
4:         Sample a batch $\{x_i\}_{i=1}^B \sim \mathcal{D}_c$
5:         $\theta \leftarrow \theta - \eta_\theta \nabla_\theta \mathcal{L}_{CL}(f_\theta; \{x_i + \delta_k\}_{i=1}^B)$
6:     **end for**
7:     **for** $t_\delta = 1$ to $T_\delta$ **do**
8:         Sample a batch $\{(x_i, y_i)\}_{i=1}^B \sim \mathcal{D}_c$
9:         **for** $t_p = 1$ to $T_p$ **do**
10:             $g_i \leftarrow \nabla_{\delta_k} \lambda \mathcal{L}_{CL}(f_\theta; \{x_i + \delta_k\}_{i=1}^B) + \mathcal{L}_{DDL}(f_\theta; \{x_i + \delta_k\}_{i=1}^B)$
11:             $\delta(y) \leftarrow \Pi_\epsilon \left( \delta(x_i) - \eta_\delta \cdot \text{sign}\left( \sum_{i:y_i=y} g_i \right) \right), \forall y$
12:         **end for**
13:     **end for**
14: **end for**
15: **Output:** Poisoned dataset $\mathcal{D}_p = \{x + \delta_k : x \in \mathcal{D}_c, k \in K\}$

---

## 4.1 SETUP

**Datasets and Models.** We conduct experiments on several benchmark datasets, including CIFAR-10/100 (Krizhevsky & Hinton, 2009), STL-10 (Coates et al., 2011), Stanford Cars (Krause et al., 2013), Oxford Flowers (Nilsback & Zisserman, 2008), Food-101 (Bossard et al., 2014), SUN397 (Xiao et al., 2010), and ImageNet (Russakovsky et al., 2015). For ImageNet, we use a subset of 100 classes, denoted as ImageNet-100. ResNet-18 serves as the default surrogate model unless stated otherwise. For target models, we consider a range of architectures, including ResNet-18/50, DenseNet-121, and VGG-19. All experiments use standard data augmentations, including resizing, random cropping, horizontal flipping, and normalization. We further evaluate our approach under two popular contrastive learning frameworks: SimCLR (Chen et al., 2020b), and BYOL (Grill et al., 2020)

**Baselines.** We compare our proposed method, DICU, with several representative approaches: Contrastive Poisoning (CP) (He et al., 2023), Unlearnable Clusters (UC) (Zhang et al., 2023), DeepConfuse (Feng et al., 2019), Synthetic Perturbations (SynPer) (Yu et al., 2022), and Adversarial Poisoning (AdvPoison) (Fowl et al., 2021a).

## 4.2 MAIN RESULTS

We summarize poisoning attacks in contrastive learning frameworks in Table 1 and compare with other baseline attacks in Table 2. Overall, our method outperforms especially in high-class-count settings. To support this observation, we evaluate on datasets with larger numbers of classes and find that the attack is significantly more effective, highlighting a fundamental vulnerability of contrastive learning in high-class settings. However, performance on Stanford Cars and Oxford Flowers is weaker, which may indicate that our attack benefits from greater variation between classes.

Table 1: Performance of indiscriminate poisoning attacks across contrastive learning algorithms and datasets. Results are reported as linear probing accuracy ($\%, \downarrow$). Clean, random noise, and classwise contrastive poisoning (CP) baselines are included for reference.

| Attack Type | CIFAR-10 | | CIFAR-100 | | ImageNet-100 |
|---|---|---|---|---|---|
| | SimCLR | BYOL | SimCLR | BYOL | SimCLR |
| None | 91.8 | 92.2 | 62.8 | 65.3 | 69.3 |
| Random Noise | 90.4 | 90.7 | 57.5 | 61.0 | 67.5 |
| CP ($\epsilon = 8/255$) | **68.0** | **56.9** | 54.6 | 37.9 | 55.6 |
| DICU ($\epsilon = 8/255$) | 79.7 | 66.3 | **36.4** | **25.0** | **2.1** |

**Impact on Downstream Task for transferability.** We poison SimCLR during training on ImageNet-100, and then evaluate the learned features by training linear classifiers on clean downstream datasets, including CIFAR-10, CIFAR-100, STL-10, and ImageNet-100 (Table 3). Despite the downstream

Table 2: Performance of our method and four baselines under high-class settings in the label-agnostic scenario. Results are reported as test accuracy (%, ↓), with the best attack highlighted in bold.

| Methods | Cars | Flowers | Food101 | Sun397 | ImageNet-100 |
|---------|------|---------|---------|--------|--------------|
| Deep-Confuse | 51.1 | 50.9 | 73.1 | 34.4 | 55.1 |
| Adv Poison | 51.9 | 50.6 | 75.1 | 38.5 | 73.8 |
| SynPar | 53.5 | 52.7 | 74.8 | 38.3 | 74.7 |
| UC | **33.6** | **35.6** | 55.3 | 20.4 | 54.8 |
| DICU | 71.8 | 69.8 | **14.6** | **14.7** | **2.1** |

Table 3: Attack transferability across Datasets and backbone architectures. Reported values are classification accuracy (%, ↓).

**Transferability across datasets**

| Attack type | Poisoning on Imagenet 100 | | | |
|-------------|---------------------------|-----------|----------|--------|
| | Imagenet-100 | CIFAR-100 | CIFAR-10 | STL-10 |
| None | 69.3 | 58.5 | 72.5 | 82.0 |
| DICU-ResNet-18 | **10.8** | **56.6** | **40.1** | **28.4** |

**Transferability across backbone architectures**

| Attack type | Poisoning on CIFAR-100 | | | |
|-------------|------------------------|-----------|--------|-------------|
| | Resnet 18 | Resnet 50 | Vgg19 | Densenet121 |
| DICU-ResNet-18 | 36.4 | 14.9 | **10.9** | 22.3 |

datasets being different from those used for poisoning, the model's performance consistently drops, demonstrating cross-dataset attacks, where poisoning on one dataset impairs performance on entirely separate datasets. We generate contrastive poisons using ResNet-18 on CIFAR-100 within the SimCLR framework. The victim model is trained on these poisoned datasets and evaluated with different architectures, including ResNet-18, ResNet-50, VGG-19, and DenseNet-121. The results indicate that increasing model complexity may not be sufficient to reduce the poisoning effect. Our attacks remain effective across all backbone architectures.

**Transferability Across different CL algorithms.** We evaluate the effectiveness of *DICU* across different contrastive learning (CL) algorithms, assessing both within-model and cross-model poisoning transferability. Table 4 reports the linear probing accuracy of SimCLR, and BYOL victim models when trained on features poisoned using *DICU*. We consider $L_\infty$- norm restriction in this experiment = 16/255. Our results show that DiCU consistently outperforms standard CP across all victim models and at-

Table 4: Cross-Model Transferability

| Attack type + attacker algorithm | Victim's algorithm | |
|----------------------------------|--------|------|
| | simclr | byol |
| *DiCP* SimCLR | 63.7 | 47.2 |
| *DiCP* BYOL | **54.8** | **44.2** |

tacker configurations. Notably, when poisons are generated using BYOL, the attack exhibits the strongest transferability–achieving the lowest accuracy on all victim models, with performance dropping as low as 44.2% for BYOL. This suggests that features poisoned under BYOL encode more generalizable divergence, making iteam task highly transferable to other CL frameworks. These findings highlight *DICU*'s ability to generalize across architectures, making it a more effective and transferable attack compared to traditional classwise contrastive poisoning.

**Defenses.** We conduct experiments on CIFAR-100 using SimCLR and ResNet-18. We ablate the hyperparameters of data augmentations, which have been studied as defenses against poisoning attacks (Tao et al., 2021; Huang et al., 2021). We test four standard augmentations: Random Noise, which adds white noise; Gauss Smooth, which applies a Gaussian filter; Cutout (DeVries & Taylor, 2017), which removes parts of the input; and MixUp (Zhang et al., 2017). We also evaluate Matrix Completion, which randomly drops pixels and reconstructs them using matrix completion (Chatterjee, 2015), and JPEG compression with quality set to 10. Table 8 shows that, except for Adversarial Training, all defenses remain stable under both attacks.

Table 5: Performance of defenses

| Methods | Accuracy |
|---------|----------|
| Clean | 58.5 |
| No defense | 36.4 |
| Random Noise (8/255) | 54.3 |
| Gaussian smooth (k=3) | 55.3 |
| MixUP | 47.8 |
| CutOut | 47.5 |
| Matrix completion | 55.7 |
| jpeg comp. | 50.5 |
| Adv. Training | 35.6 |

Table 7: Accuracy under different reference direction strategies for indiscriminate poisoning.

| setup | Reference Direction Strategy | Accuracy (%) |
|-------|------------------------------|--------------|
| clean | NA | 91.8 |
| D-1 | Random cosine values between -1 and 1 | 63.7 |
| D-2 | Equispaced cosine values between -1 and 1 | **58.7** |
| D-3 | Manually assigned angles per class | 63.4 |

## 4.3 ABLATION STUDY

We perform an ablation study on CIFAR-100 using SimCLR to evaluate the impact of different loss components on DiCU's attack efficacy. Table 6 shows that only the DDL term alone can effectively perturb the loss, while combining it with CP or using CP alone is less effective. We study how directional reference values affect attack effectiveness by defining three divergence configurations: D-1, D-2, and D-3 (Table 4). D-1 uses 10 random cosine values, D-2 uses uniformly spaced values, and D-3 assigns specific angles per class to cover different quadrants. These settings control the distribution of poisoned samples on the unit hypersphere. Using SimCLR on CIFAR-10, we evaluate the impact of each strategy on learned representations. D-2, with evenly spaced angles, consistently achieves the lowest downstream accuracy, showing that uniform angular separation maximizes semantic disruption. These results emphasize the role of directional divergence in poisoning attacks.

Table 6: Ablation on loss components.

| Loss Terms | Accuracy |
|------------|----------|
| CP | 54.6 |
| CP+DDL | 36.4 |
| DDL | 17.8 |

## 4.4 ANALYZING PROPERTIES AND LOSS DYNAMICS

**Loss Dynamics During DiCU Optimization.** To better understand the training behavior of our proposed attack, we track both the contrastive loss and the directional divergence loss across epochs. As shown in Figure 2a, the contrastive loss steadily declines, indicating that the model continues to optimize its alignment and uniformity objectives, even when poisoned data is used. In parallel, the divergence loss shown in Figure 2b rises early and then stabilizes, suggesting that the imposed divergence constraint remains consistently active during training. This sustained directional enforcement drives poisoned and clean views apart at the representation level, fragmenting intra-class structure and weakening semantic consistency across the embedding space.

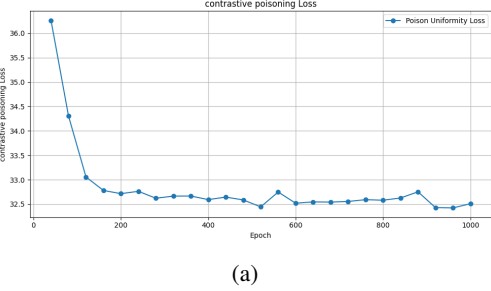

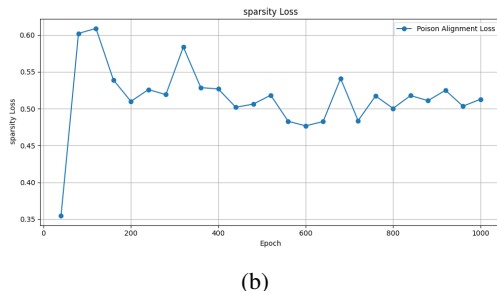

(a)                                                          (b)

Figure 2: Training losses under sparsity-aware contrastive poisoning. (a) Contrastive loss decreases steadily. (b) Sparsity loss remains consistently high, enforcing intra-cluster dispersion.

**Behavioral Analysis of Contrastive Learning Under Poisoned Data.** Unlike classification losses, contrastive learning (CL) objectives such as *InfoNCE* aim to produce robust representations by optimizing both *alignment*-bringing augmented views of the same instance closer-and *uniformity*-dispersing representations across the feature space (Wang & Isola, 2020). To evaluate how *DiCU* interferes with these objectives, we track their behavior during training. As shown in Figure 3, we visualize the alignment and uniformity trends for SimCLR on CIFAR-10 under our proposed *DiCU*. Blue curves indicate poisoned samples, while orange curves represent clean samples. Notably, the alignment loss shows a large discrepancy, suggesting that *DiCP* primarily targets and disrupts alignment. This behavior is further supported by the cosine similarity distributions in Figure 3.

Poisoned view pairs remain highly aligned with cosine distances near zero, while clean view pairs show broader variance, indicating weakened alignment. Despite appearing well-aligned, the model fails to learn semantically meaningful representations for poisoned samples-effectively deceiving the alignment objective.

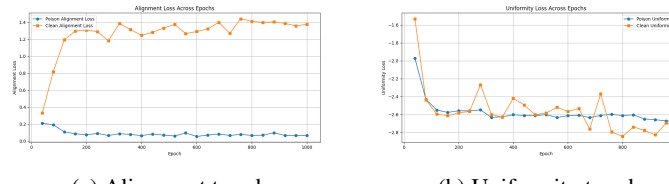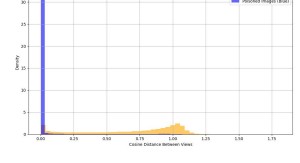

(a) Alignment trend      (b) Uniformity trend      (c) Distri. of cosine similarity

Figure 3: (a, b) Evolution of alignment and uniformity losses over training epochs. (c) Cosine distance distributions between embeddings of two augmented views for poisoned (blue) and clean (orange) samples.

## 4.5 VISUALIZING ATTACKER AND VICTIM PERSPECTIVES ON POISONED FEATURES

To better understand how our poisoning attack alters representation learning, we visualize the feature space using t-SNE plots from both the attacker (Figure 5) and victim (Figure 4) perspectives. From the attacker's side, *K-means* clustering reveals clean geometric separation, yet these clusters exhibit strong semantic misalignment when overlaid with true class labels—indicating the attack preserves spatial structure while fragmenting class identity. This mismatch is further supported by heatmaps showing multiple classes collapsing into a small subset of clusters, revealing a loss of class diversity. In contrast, t-SNE plots on the victim side reveal heavily entangled class labels with no coherent cluster boundaries. Despite apparent groupings, the representations lack class specificity, confirming that the poisoning effectively dismantles semantic structure while maintaining a deceptive appearance of order. This divergence between geometric and semantic organization highlights the stealthy nature of our attack: the learned features appear structured, but are fundamentally misaligned with the underlying data distribution—thereby degrading downstream performance while evading simple detection. We present a comparison with contrastive poisoning in Figure 6, where it is evident that, on the victim side, contrastive poisoning disrupts semantic alignment while also degrading the geometric structure of the learned representations.

## 4.6 QUALITATIVE ANALYSIS OF LEARNED PERTURBATIONS

Figure 7 visualizes the learned perturbations $\delta$ produced by *DICU* when attacking SimCLR on CIFAR-10. Noise components exhibit consistent, structured patterns, indicating that the optimization process captures meaningful directions in representation space. Clean and poisoned image pairs are also shown to demonstrate that the perturbations remain visually imperceptible, yet still disrupt semantic alignment effectively.

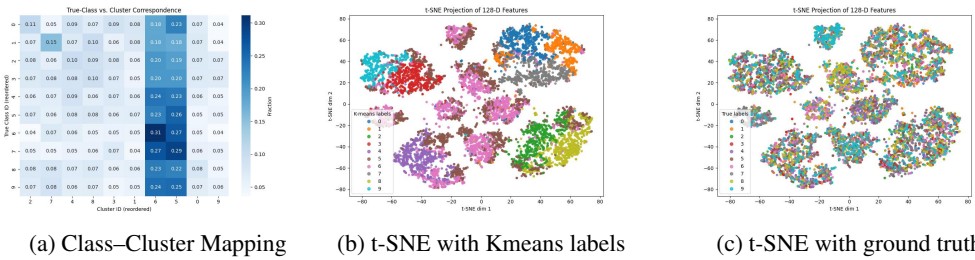

(a) Class–Cluster Mapping      (b) t-SNE with Kmeans labels      (c) t-SNE with ground truth

Figure 4: Visualizing *victim* perspectives on poisoned features.(a) Class–cluster heatmap shows collapse into a few dominant clusters. (b) t-SNE with K-means labels reveals structured but semantically misaligned clusters. (c) t-SNE with ground-truth labels shows strong class mixing and loss of semantic structure.

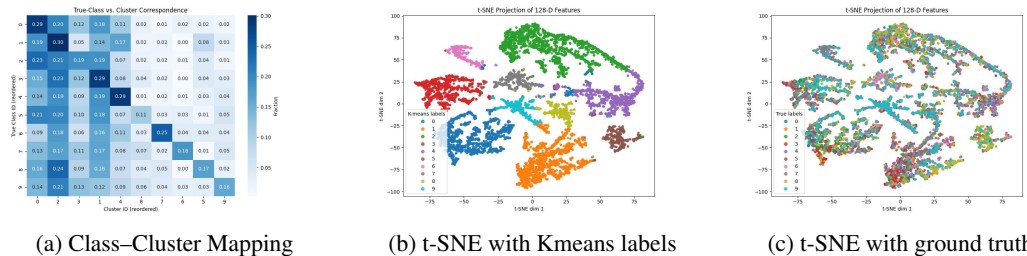

(a) Class–Cluster Mapping    (b) t-SNE with Kmeans labels    (c) t-SNE with ground truth

Figure 5: Visualizing *attacker* perspectives on poisoned features. (a) Heatmap shows partial alignment between clusters and true classes, with noticeable class leakage. (b) t-SNE with K-means labels shows clean cluster geometry but loss of semantic alignment. (c) t-SNE with true labels reveals severe class mixing, confirming disruption of class separability.

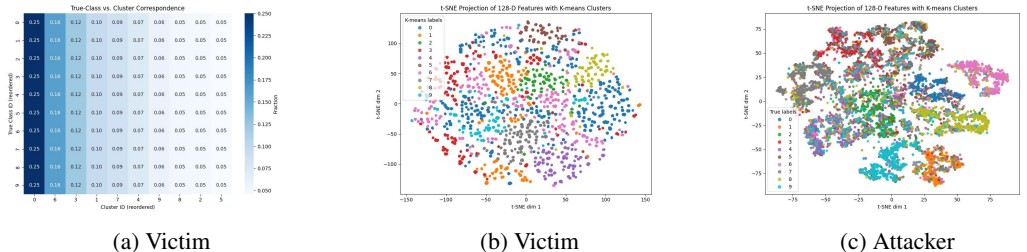

(a) Victim    (b) Victim    (c) Attacker

Figure 6: Visualizing Victim and Attacker Perspectives on Poisoned Features under Contrastive Poisoning (He et al., 2023) (a) Class–Cluster Mapping. (b)-(c) t-SNE with Kmeans labels.

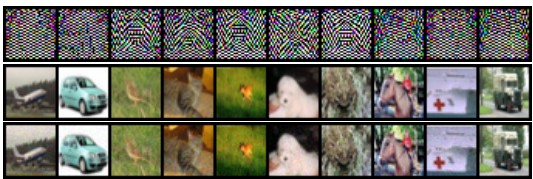

Figure 7: Visualization of poisoning noise in SimCLR trained on CIFAR-10. The first row display the learned $\delta$ perturbations. The second and third rows display clean images and their corresponding poisoned versions for each CIFAR-10 class.

## 5 CONCLUSION

We present our first investigation into intra-class sparseness induced by external forces, which increases the difficulty of unlearning. We find that this difficulty grows in large-class scenarios, as the closeness of different classes in the embedding space makes attacks easier when class sparsity increases. This leads to the mixing of embeddings from nearby classes while retaining geometric semantics, resulting in stealthy attacks. Learned poisons are also transferable across datasets and architectures, and increasing model complexity alone is insufficient to defend against them. However, certain data augmentation–based defenses provide robust protection, whereas adversarial training offers limited defense against our method.

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

# A  APPENDIX

## A.1  PRELIMINARIES

### A.1.1  CONTRASTIVE LEARNING

Let the input space be denoted as $\mathcal{X}$ and the representation (embedding) space as $\mathbb{R}^d$. A contrastive learning model is parameterized by an encoder $f_\theta : \mathcal{X} \to \mathbb{R}^d$, which maps an input sample $x \in \mathcal{X}$ to its representation $z = f_\theta(x)$. We assume that all representations are normalized to lie on the unit hypersphere $\mathbb{S}^{d-1}$.

**Alignment.** Contrastive learning seeks to minimize the distance between positive pairs, i.e., augmentations of the same input $x$. Formally, for a pair $(x_i, x_i^+)$, the alignment objective is given by:

$$\mathcal{L}_{\text{align}} = \mathbb{E}_{(x_i, x_i^+)} \big[ \| f_\theta(x_i) - f_\theta(x_i^+) \|_2^2 \big]. \tag{7}$$

A small alignment loss indicates that representations of augmented views of the same sample are closely aligned in the embedding space.

**Uniformity.** To avoid representational collapse, contrastive learning enforces uniform coverage of the hypersphere. Uniformity can be quantified as:

$$\mathcal{L}_{\text{unif}} = \log \mathbb{E}_{x_i, x_j \sim \mathcal{D}} \big[ e^{-t \| f_\theta(x_i) - f_\theta(x_j) \|_2^2} \big], \tag{8}$$

where $t > 0$ is a temperature hyperparameter. Low $\mathcal{L}_{\text{unif}}$ implies that embeddings are well-dispersed across the representation space.

### A.1.2  CONTRASTIVE POISONING (CP)

We briefly review the key components of contrastive poisoning as introduced in prior work (He et al., 2023), including poison generation, data augmentation, and dual-branch gradient propagation. As these elements form the foundation of our method, we adopt and build upon them in this work.

### A.1.3  CONTRASTIVE POISONING GENERATION

Contrastive Poisoning (CP) is an indiscriminate data poisoning strategy designed to undermine the ability of contrastive learning (CL) algorithms to learn meaningful representations from training data. Instead of targeting specific classes or examples, CP introduces perturbations broadly across the dataset to degrade representation quality in a self-supervised setting. In the standard CL pipeline (e.g., SimCLR, MoCo, BYOL), a feature encoder and a projection head are jointly trained to align views of the same input (positive pair) while pushing apart views of different inputs (negative pair). The goal of CP is to generate imperceptible perturbations that interfere with this alignment process, misleading the model to minimize the CL objective while failing to capture semantic structure.

To accomplish this, the attacker selects a target CL method and jointly optimizes the feature encoder parameters $\theta$ and an input-specific perturbation function $\delta(x)$. The poisoning objective is formulated as:

$$\min_{\theta, \delta : \| \delta(x) \|_\infty \leq \epsilon} \mathbb{E}_{\{x_i\}_{i=1}^B \sim \mathcal{D}_c} \, \mathcal{L}_{\text{CL}} \big( f_\theta; \{x_i + \delta(x_i)\}_{i=1}^B \big), \tag{9}$$

where $\mathcal{L}_{\text{CL}}$ denotes a contrastive loss (e.g., InfoNCE), $B$ denotes the batch size and $\mathcal{D}_c$ is the clean dataset. Optimization proceeds in alternating steps: the encoder is updated via stochastic gradient

descent (SGD), while the perturbations $\delta(x)$ are refined using projected gradient descent (PGD) under an $\ell_\infty$ constraint to ensure imperceptibility.

This formulation lays the foundation for our work, which extends CP by introducing directional divergence constraints that target intra-class structure in the learned representation space.

### A.1.4 DUAL BRANCH POISON PROPAGATION

In supervised learning, optimizing data perturbations for poisoning is relatively straightforward, as gradients can be directly obtained through the cross-entropy loss, i.e., $\nabla_x \mathcal{L}_{\text{CE}}(h(x), y)$. However, contrastive learning presents additional complexity: the loss function depends on the relationship between multiple data points in a batch, and many frameworks–such as MoCo and BYOL–incorporate a momentum encoder that is detached from the main training pipeline.

In these settings, the momentum encoder is updated via an exponential moving average (EMA) of the online encoder, and it does not participate in backpropagation by default. As a result, conventional poisoning approaches only use gradients from the online encoder–this is known as the *single-branch* gradient flow.

Following the approach of He et al. (He et al., 2023), we adopt a *dual-branch* gradient scheme, where gradients are propagated through both the online encoder and the momentum encoder during poison optimization. This modification provides richer gradient signals and enables more effective perturbation updates, especially in momentum-based frameworks like MoCo and BYOL. As demonstrated in prior work, dual-branch gradient flow significantly improves the quality of learned poisons compared to the single-branch variant.

### A.1.5 DATA AUGMENTATION

Contrastive learning heavily relies on strong data augmentations (e.g., cropping, color jittering, brightness shifts), which are essential for learning invariant representations. However, these augmentations pose a challenge for poisoning attacks: if not properly incorporated into the optimization loop, they can neutralize the effect of perturbations by altering the poisoned inputs before the encoder processes them. To ensure the effectiveness of poisoning in the presence of augmentations, we follow he et al.

### A.2 DECLARATION OF LLM USAGE

Large Language Models (LLMs) were not involved in the core methodology or experiments of this research. Any language editing assistance (e.g., improving phrasing or clarity) did not affect the scientific contributions or the originality of the work.

