# OpenReview forum: "Divergence-Induced Contrastive Unlearning via Directed Representation Shifts"
_ICLR.cc/2026/Conference — ICLR 2026 Conference Withdrawn Submission_

### Official Review · Reviewer_tW2F · 2025-10-28

**Soundness:** 2
**Presentation:** 3
**Contribution:** 2
**Rating:** 4
**Confidence:** 3

**Summary:**

This paper proposes a novel method for generating Unlearnable Examples, termed Divergence-Induced Contrastive Unlearning (DICU). DICU aims to make data unusable for CL by introducing imperceptible perturbations designed to disrupt the learned representation space. Unlike prior work focusing on supervised settings or global confusion, DICU specifically targets the intra-class structure in CL by introducing a direction-aware divergence regularization term alongside a contrastive poisoning objective. The method operates in two phases: contrastive poisoning and divergence induction. The authors claim DICU is particularly effective in settings with a high number of classes, creating stealthy and robust perturbations that significantly degrade downstream task performance.

**Strengths:**

This paper introduces a novel poisoning strategy specifically for CL, focusing on manipulating intra-class structure through direction-aware divergence regularization. This contrasts with previous methods and offers a new perspective on generating UEs in self-supervised settings.

**Weaknesses:**

1. While motivated by geometric intuition, the theoretical link between the proposed divergence mechanism and the resulting degradation in downstream task performance could be more rigorously established. The geometric propositions mentioned in Section 3 are not fully developed or proven within the paper.

2.Computational Cost: The algorithm 1 for generating perturbations likely incurs significant computational overhead. The paper lacks discussion on the efficiency of DICU perturbation generation compared to baseline methods.

3. DICU introduces new components and hyperparameters, such as the reference direction strategy ($\phi_{ref, y_i}$) and the balancing weight $\lambda$. The ablation study on reference directions is useful, but a broader sensitivity analysis regarding these choices is missing.

4.  Lack of comparison for more methods and references. Such as [1].

[1]: Efficient availability attacks against supervised and contrastive learning simultaneously.

**Questions:**

1. Could you investigate further why DICU performs less effectively on datasets like Cars and Flowers?

2. Please include details on the computational cost.

3. Could you discuss potential stronger defense strategies targeting DICU's mechanism?

4. Please provide a sensitivity analysis for key hyperparameters, especially $\lambda$ and the parameters defining the reference direction strategy. How robust is the method to these choices?

---

### Official Review · Reviewer_TaPn · 2025-10-30

**Soundness:** 3
**Presentation:** 2
**Contribution:** 2
**Rating:** 2
**Confidence:** 5

**Summary:**

The paper studies poisoning-based unlearnable examples for contrastive learning and proposes Divergence-Induced Contrastive Unlearning (DICU), which injects direction-aware divergence into the poisoning objective to fragment intra-class manifolds and encourage cross-class mixing. The authors analyze representation alignment/uniformity under poisoning, formalize a two-phase attack (contrastive poisoning + divergence induction), and report strong drops in linear-probe accuracy—particularly in high-class-count settings—across SimCLR/BYOL and multiple datasets.

**Strengths:**

1. The core contribution, "Direction-Aware Divergence Regularization," is original. Unlike prior work (like CP) that primarily focuses on manipulating the InfoNCE loss, this paper proposes a novel, geometry-based attack mechanism: systematically destroying the cluster structure by actively increasing and guiding intra-class sparsity. This is a creative attack strategy that differs from existing approaches.
2. Although there is some vague phrasing in the introduction, the paper's core methodology (Section 3) and algorithm (Algorithm 1) are described with relative clarity.

**Weaknesses:**

1. The authors vaguely attribute this failure in the text to "benefits from greater variation between classes," which is insufficient. The authors must provide a deeper analysis or supplementary experiments to investigate: why does the "increase intra-class sparsity" mechanism fail so completely on fine-grained tasks? Is it because the intra-class variance in these tasks is inherently high, causing the mechanism to backfire?
2. The critique of prior work (CP) in the Introduction is logically flawed. The authors first state that CP causes "clean views to move apart," but then immediately claim it "does not address interactions between clean and poisoned pairs," which seems contradictory. Furthermore, the phrase "why the attack is so vulnerable" is extremely ambiguous: is the CP attack itself "vulnerable," or is the model "vulnerable" under the CP attack?
3. The review of existing studies is insufficient, lacking coverage of recent papers on Unlearnable Examples published in the past two years.
4. Evaluated defenses are primarily data augmentations and simple transforms; adversarial training is included but limited. Consider stronger or certified defenses, feature denoising, and recent CL-specific robust training.

**Questions:**

1. Is the "intra-class divergence" mechanism fundamentally unsuitable for fine-grained tasks with high intra-class variance and low inter-class variance?
2. Partial poisoning: How does attack efficacy scale when only p% of training samples are perturbed (e.g., p ∈ {10, 30, 50, 70})? Does the divergence mechanism still dominate under low coverage?
3. Could you evaluate stronger defenses (e.g., adversarial/consistency-regularized CL, feature denoising, robust contrastive objectives) or explain why they are expected to fail?
4. You note weaker results on Cars/Flowers. Can you analyze class-count vs. intra-class variation to explain when DICU is most/least potent and how to adapt it?

---

### Official Review · Reviewer_jZ37 · 2025-10-31

**Soundness:** 2
**Presentation:** 3
**Contribution:** 2
**Rating:** 4
**Confidence:** 4

**Summary:**

The paper proposes Divergence-Induced Contrastive Unlearning (DiCU), an indiscriminate poisoning framework for contrastive self-supervised learning. The core idea is to fragment intra-class structure via directional angular divergence between clean and poisoned views, formalized by two geometric definitions and instantiated with a bi-level optimization routine. Experiments on SimCLR/BYOL using CIFAR-10/100, ImageNet-100, and several fine-grained datasets report large linear-probe accuracy drops, plus transferability and defense studies.

**Strengths:**

- The paper targets intra-class fragmentation through explicit angular constraints and two formal definitions, which is a interesting geometric formulation of a poisoning objective.

- The empirical section spans multiple datasets and includes ablations on loss components and reference-direction strategies, with uniformity tracking and t-SNE visualizations.

- The paper is well organized and easy to read.

**Weaknesses:**

- Limited novelty relative to existing contrastive-poisoning and cluster-level unlearning. The paper positions DiCU as a principled advancement, yet the mechanism largely augments prior contrastive poisoning (CP) with an angular divergence regularizer and staging. DiCU’s directional divergence appears as a straightforward geometric variant without theoretical guarantees that it achieves qualitatively new capabilities beyond contrastive poisoning.

- The bi-level optimization lacks sufficient theoretical support. The paper claims to employ a bi-level optimization framework. However, the inner and outer objectives are not clearly distinguished, and the method relies solely on gradient-based updates without convergence analysis. Since first-order gradient-based bi-level optimization is difficult to guarantee convergence for in non-convex cases, the absence of theoretical or empirical exploration weakens the methodological soundness of the work.

- The text claims that DiCU is especially effective in high-class-count settings, but Table 2 shows DiCU performs poorly on fine-grained Cars and Flowers, while excelling on other datasets. The paper acknowledges weaker performance but provides no diagnosis. This inconsistency requires a principled analysis.

**Questions:**

See weakness

---

### Official Review · Reviewer_jX9B · 2025-10-31

**Soundness:** 3
**Presentation:** 2
**Contribution:** 1
**Rating:** 2
**Confidence:** 4

**Summary:**

This paper introduces the direction-aware divergence regularization into the contrastive poisoning framework to create unlearnable datasets against contrastive learning.

**Strengths:**

1. The proposed method is motivated.
2. Experiments are conducted on multiple datasets.

**Weaknesses:**

1. This paper seriously misuses the term unlearning. Machine Unlearning is an independent research area that has attracted significant attention in both classification and generation tasks. It refers to removing the memory of certain training data from a trained model. However, this paper actually studies unlearnable samples designed for data protection. Such misuse of terminology should not occur.
2. A lot of related literature in the field of unlearnable examples has been missed. Surprisingly, all the cited references are from 2023 and earlier.
3. The performance should be compared to two missed but important baseline methods [1, 2].

	[1] Ren J, Xu H, Wan Y, et al. Transferable Unlearnable Examples. ICLR. 2023.

	[2] Wang Y, Zhu Y, Gao X S. Efficient availability attacks against supervised and contrastive learning simultaneously. NeurIPS. 2024.
4. DICU does not show a consistent advantage compared to UC/CP.

**Questions:**

1. What is DiCP in Table 4?
2. Since Table 6 shows that using DDL alone yields the best performance, why is it still combined with CP loss in the proposed algorithm?
3. What if the poisoned dataset is used for supervised learning?

---

### Note · Authors · 2025-11-14

I have read and agree with the venue's withdrawal policy on behalf of myself and my co-authors.